# Exome Survey and Candidate Gene Re-Sequencing Identifies Novel Exstrophy Candidate Genes and Implicates *LZTR1* in Disease Formation

**DOI:** 10.3390/biom13071117

**Published:** 2023-07-13

**Authors:** Ricarda Köllges, Jil Stegmann, Sophia Schneider, Lea Waffenschmidt, Julia Fazaal, Katinka Breuer, Alina C. Hilger, Gabriel C. Dworschak, Enrico Mingardo, Wolfgang Rösch, Aybike Hofmann, Claudia Neissner, Anne-Karolin Ebert, Raimund Stein, Nina Younsi, Karin Hirsch-Koch, Eberhard Schmiedeke, Nadine Zwink, Ekkehart Jenetzky, Holger Thiele, Kerstin U. Ludwig, Heiko Reutter

**Affiliations:** 1Institute of Human Genetics, University of Bonn, 53127 Bonn, Germany; ricarda.koellges@posteo.de (R.K.);; 2Institute of Anatomy and Cell Biology, Medical Faculty, University of Bonn, 53127 Bonn, Germany; 3Department of Pediatrics and Adolescent Medicine, University Hospital Erlangen, 91054 Erlangen, Germany; 4Department of Neuropediatrics, University Hospital Bonn, 53127 Bonn, Germany; 5Department of Pediatric Urology, Clinic St. Hedwig, University Medical Center Regensburg, 93053 Regensburg, Germany; 6Department of Urology and Pediatric Urology, University Hospital Ulm, 89081 Ulm, Germany; 7Center for Pediatric, Adolescent and Reconstructive Urology, University Medical Center Mannheim, University Heidelberg, 69117 Mannheim, Germany; 8Division of Pediatric Urology, Department of Urology, University Hospital Erlangen, 91054 Erlangen, Germany; 9Clinic for Pediatric Surgery and Pediatric Urology, Klinikum Bremen-Mitte, 28205 Bremen, Germany; 10Department of Child and Adolescent Psychiatry, University Medical Center of the Johannes Gutenberg University Mainz, 55131 Mainz, Germany; 11Cologne Center for Genomics, University of Cologne, 50923 Cologne, Germany; 12Division of Neonatology and Pediatric Intensive Care, Department of Pediatrics and Adolescent Medicine, University Hospital Erlangen, 91054 Erlangen, Germany

**Keywords:** exome analysis, molecular inversion probe, exstrophy, cloacal exstrophy

## Abstract

Background: The bladder exstrophy-epispadias complex (BEEC) is a spectrum of congenital abnormalities that involves the abdominal wall, the bony pelvis, the urinary tract, the external genitalia, and, in severe cases, the gastrointestinal tract as well. Methods: Herein, we performed an exome analysis of case-parent trios with cloacal exstrophy (CE), the most severe form of the BEEC. Furthermore, we surveyed the exome of a sib-pair presenting with classic bladder exstrophy (CBE) and epispadias (E) only. Moreover, we performed large-scale re-sequencing of CBE individuals for novel candidate genes that were derived from the current exome analysis, as well as for previously reported candidate genes within the CBE phenocritical region, 22q11.2. Results: The exome survey in the CE case-parent trios identified two candidate genes harboring de novo variants (*NR1H2*, *GKAP1*), four candidate genes with autosomal-recessive biallelic variants (*AKR1B10*, *CLSTN3*, *NDST4*, *PLEKHB1*) and one candidate gene with suggestive uniparental disomy (*SVEP1*). However, re-sequencing did not identify any additional variant carriers in these candidate genes. Analysis of the affected sib-pair revealed no candidate gene. Re-sequencing of the genes within the 22q11.2 CBE phenocritical region identified two highly conserved frameshift variants that led to early termination in two independent CBE males, in *LZTR1* (c.978_985del, p.Ser327fster6) and in *SLC7A4* (c.1087delC, p.Arg363fster68). Conclusions: According to previous studies, our study further implicates *LZTR1* in CBE formation. Exome analysis-derived candidate genes from CE individuals may not represent a frequent indicator for other BEEC phenotypes and warrant molecular analysis before their involvement in disease formation can be assumed.

## 1. Introduction

The bladder exstrophy-epispadias complex (BEEC; OMIM %600057) characterizes a spectrum of human congenital anomalies comprising malformations of the urinary tract and the genitalia, pelvis, abdominal wall, and, occasionally, the spine and gastrointestinal tract [1,2]. The BEEC encompasses a vast severity spectrum incorporating different phenotypes, including epispadias (E) as the mildest phenotype, classic bladder exstrophy (CBE) as the intermediate and most common form, and cloacal exstrophy (CE) as the most severe phenotype [3]. CE is also referred to as OEIS (omphalocele, exstrophy, imperforate anus, and spinal defects) complex [4]. Epidemiological studies state the incidence rates at about 2.4:100,000 births for E, 1–2:50,000 births for CBE, and 0.5–1:200,000 births for CE [1], with an overall birth prevalence of 1:10,000 in children of European descent [3]. A male-to-female ratio ranging from 1.5:1 to 6:1 [5] is reported, with CBE being more frequent in males and CE being more common in females [6]. Additional anomalies of the urinary tract, such as an ectopic kidney, a horseshoe kidney, renal hypo- or agenesis, and ureteropelvic junction obstruction are present in one-third of all cases, mainly in the form of the CE phenotype [2,5]. Both sexes are affected by impaired sexual function and fertility issues [2,7,8,9]. Fertility in men is decreased due to low ejaculation volumes and poor sperm quality [8,9].

Multiple findings suggest that genetic factors play an important role in BEEC etiology: (i) increased recurrence risk for the siblings of CBE individuals [5,10,11], (ii) increased recurrence risk for the offspring of affected individuals [11], (iii) a higher concordance rates among monozygotic compared to dizygotic twins [12], and (iv) the report of several multiplex families in the literature [13]. Previously, candidate genes for monogenic forms were identified through array-based molecular karyotyping [14] and exome analysis [3]. Using exome analysis, we recently identified *SLC20A1* as a monoallelic candidate gene for CE [15]. To identify further candidate genes, we performed exome analysis in 14 CE case-parent trios and in one affected sib-pair (CBE and epispadias only). To prioritize the identified candidate genes, we re-sequenced 480 BEEC individuals for the prioritized candidate genes. Furthermore, we re-sequenced previously reported candidate genes within the CBE phenocritical region, 22q11.2 [14]. Based on our findings, we suggest that the gene *LZTR1* is involved in CBE formation.

## 2. Materials and Methods

### 2.1. Individuals

Exome analysis was performed in 14 CE case-parent trios and 1 affected sib-pair presenting with CBE and epispadias only. The complete re-sequencing cohort comprised 480 BEEC individuals (310 males and 169 females; in 1 case, the gender was unknown). Ethical consent was obtained by the Ethics Committee of the Medical Faculty of the University of Bonn (Lfd.Nr.031/19). Written informed consent was provided by all participating families prior to the study.

### 2.2. DNA Preparation and Exome Sequencing

The DNA of individuals and their parents was extracted from saliva samples, using the Oragene DNA Kit (DNA Genotek Inc., Ottawa, ON, Canada), or from blood samples using the Chemagic Magnetic Separation Module I (Cheagen, Baesweiler, Germany). In the first step, exome sequencing was performed for 14 CE case-parent trios and 1 affected CBE and E sib-pair at the Next-Generation Sequencing Laboratory of the Cologne Center for Genomics (CCG), using the Agilent SureSelect Human All Exon V6 for 12 families, the NimbleGen SeqCap EZ Human Exome Library v 2.0 for 2 families, and the Agilent SureSelect All Exon V7 for 1 family. After validation (2200 TapeStation; Agilent Technologies, Santa Clara, CA, USA) and quantification (Qubit System; Invitrogen, Waltham, MA, USA), pools of libraries were generated and subsequently sequenced on the Illumina HiSeq 4000 sequencing instrument, using a paired-end 2 × 100 bp protocol. The mean coverage of the presented exome data was 70,933 reads. In total, 92.31% of the targeted bases were covered by at least 20×. Since we filtered the reads with a minimum coverage of 10×, all the prioritized variants were validated using Sanger sequencing. Data analysis and filtering of the mapped target sequences were accomplished using the “Varbank” exome and genome analysis pipeline, version 2.0 (https://varbank.ccg.uni-koeln.de, accessed on 1 May 2000).

### 2.3. Filtering

The exome data were filtered for different inheritance patterns (autosomal-dominant variants, autosomal-recessive genes with homozygous and compound heterozygous variants, and uniparental-disomy disease variants) and underwent a visual quality-control check with Varbank2. We filtered all case-parent trios for de novo mutational events and autosomal-recessive disease variants. In the case of a dominant (de novo) disease model, we only considered those variants with a minor allele frequency (MAF) in gnomAD [16] of <0.00001. We excluded low-quality variants within the targeted regions and the flanking 100 bp. In the case of a recessive disease model, we only considered those variants with a MAF in gnomAD of ≤0.01. Filtering further included the following algorithm, involving: (a) variants present in gnomAD [16] v.2.1; (b) GnomAD [16] (the number of “loss of function” mutations with “homozygous individuals” and the number of “missense” mutations with homozygous individuals); (c) conservation of amino acids (aa) across the species Hs (*Homo sapiens*), Mm (*Mus musculus*), Gg (*Gallus gallus*, Xt (*Xenopus tropicalis*), Dr (*Danio rerio*) [17]; (d) conservation of bases across species (Hs, Mm, Gg, Xt, Dr) [17]; (e) prediction tools (PolyPhen-2 [18], SIFT [19], CADD [20]), (see Table A1 in Appendix B), (f) the presence of described knockout animal models (Mm, Dr) [21,22]; (g) the presence of expression data (https://proteinatlas.org, accessed on 1 May 2000) [23]; (h) the affection of functional protein domains [24]; (i) STRING (computational predicted interaction) [25]; (j) OMIM (the gene–phenotype relationship; https://omim.org/, accessed on 1 May 2000); (k) entries in GeneMatcher [26]; (l) entries in Phenoscanner [27,28]; (m) entries in ClinVar [29]; (n) entries in PubMed [25].

### 2.4. Molecular Inversion Probe (MIP) Assay and Sanger Validation

In the second step, prioritized candidate genes were re-sequenced using a “Molecular Inversion Probe (MIP) Assay” for the entire BEEC cohort. We included the 7 candidate genes prioritized in our exome analysis, along with 7 candidate genes that were previously described in the 22q11.2 CBE phenocritical region [14,30,31]. To cover all 192 protein-coding-transcripts of these 14 candidate genes, 335 MIPs were designed with amplicon lengths of between 165 and 189 bp (see Appendix A), using an in-house version of the MIPgen tool [32]. Three balancing runs and one re-balancing run of the MIPs were performed using the MiSeq^®^ with Reagent Kit v2 (Illumina, San Diego, CA, USA). The final pooled MIP libraries were then sequenced with the NovaSeq 6000^®^ SP XP-Workflow Reagent Kit v1.5 (300 cycles), using 2 × 125 bp. A Q30-Score of 82.93% was reached. The sequencing identified 1252 variants, of which only those variants that were covered by more than 10 reads were considered. To prioritize these variants, we applied the filter algorithm described above. Final validation of the remaining variants found in individuals and their parents was performed via Sanger sequencing with an ABI 3730 XL DNA analyzer (Life Technologies/Thermofisher, Schwerte, Germany) by Azenta Life Science.

## 3. Results

### 3.1. Exome Analysis

Filtering of the CE case-parent exome data identified seven candidate genes. These genes comprised two candidate genes with autosomal-dominant monoallelic de novo variants (*NR1H2*, *GKAP1*), three candidate genes with autosomal-recessive biallelic compound heterozygous variants (*CLSTN3*, *AKR1B10*, *NDST4*), one candidate gene with an autosomal-recessive biallelic homozygous variant (*PLEKHB1*), and one candidate gene with suggestive uniparental disomy (*SVEP1*). All variants were validated by Sanger sequencing. Exome analysis of one affected sib-pair did not identify any plausible variants/candidate genes.

#### 3.1.1. Autosomal-Dominant Candidate Genes (*GKAP1*, *NR1H2*)

##### *GKAP1* (G KINASE-ANCHORING PROTEIN; OMIM *611356)

Individual 420_501 was found to carry a novel de novo heterozygous missense variant, c.737A>C (p.Gln246Pro), in exon 8 (Ensembl GRChr37/hg19 transcript ENST00000376371.7) of *GKAP1*. However, SpliceAI predicts a high chance of donor loss for this variant so it might not be a missense but, instead, a splicing variant.

The variant was predicted to be deleterious in SIFT and benign in PolyPhen-2, with a CADD13-PHRED score of 26.1. The aa is evolutionarily conserved down to Mm and Gg (Gln), while Xt and Dr show unalienable bases in the gap region. Our in-house murine transcriptome database of relevant uro-rectal tissue shows expression over the whole time period, with an upregulation in E15.5: log2FoldChange (15.5) of 0.99659903 and a *p*-value (15.5) of 3.1474 × 10^−8^, with a baseMean (15.5) of 949.642679. *GKAP1* has not been associated with any human disease phenotype so far (Table 1).

##### *NR1H2* (NUCLEAR RECEPTOR SUBFAMILY 1, GROUP H, MEMBER 2; OMIM *600380)

Individual 420_501 was found to also carry a novel de novo heterozygous missense variant c.718C>T (p.Arg240Cys) in exon 6 (Ensembl GRChr37/hg19 transcript ENST00000253727.10) of *NR1H2*. The variant was predicted to be deleterious in SIFT and probably damaging in PolyPhen-2, with a CADD13-PHRED score of 28.5. The aa is highly conserved down to Mm, Gg, Xt, and Dr. Our in-house murine transcriptome database did not show any expression of this variant during the embryonic stages of E10.5, E.12.5, and E15.5. *NR1H2* has not been associated with any human disease phenotype so far (Table 1).

#### 3.1.2. Autosomal-Recessive Candidate Gene with Homozygous Variants (*PLEKHB1*)

##### *PLEKHB1* (PLECKSTRIN HOMOLOGY DOMAIN_CONTAINING PROTEIN; FAMILY B; MEMBER 1; OMIM *607651)

We identified one autosomal-recessive candidate gene, *PLEKHB1*, for which individual 420_501 carried a homozygous missense variant, c.76G>A (p.Gly26Ser) in exon 2 of 8 (Ensembl GRChr37/hg19 transcript ENST00000354190.10) (Table 2). It was predicted to be deleterious by SIFT and probably damaging by PolyPhen-2, with a CADD13_PHRED score of 28.4. The aa is conserved to Mm. Our in-house murine transcriptome database shows the continuous expression of this variant during the embryonic stages E10.5, E.12.5, and E15.5. The protein has only one annotated domain (aa21-128) and the variant lies at the beginning of this domain. *PLEKHB1* has not been associated with any disease phenotype so far.

#### 3.1.3. Autosomal-Recessive Candidate Genes with Compound Heterozygous Variants (*CLSTN3*, *AKR1B10*, and *NDST4*)

##### *CLSTN3* (CALSYNTENIN; OMIM *611324)

Individual 390_501 carried two compound heterozygous missense variants, c.2285A>T (p.Gln762Leu) and c.2626C>T (p.Arg876Cys), in exons 15 and 17 of 18 (Ensembl GRChr37/hg19 transcript ENST00000266546.11) of *CLSTN3* (Table 3). The variants are predicted to be tolerated and deleterious in SIFT, benign and possibly damaging in PolyPhen-2, and have a CADD13_PHRED score of 20.2 and 25.3, respectively. The bases are conserved over Mm and Gg (A/C). Xt and Dr show base variations for chr12:7303179 and the base is conserved for Dr at chr12:7310183. Our in-house murine transcriptome database shows the continuous expression of *Clstn3* during the embryonic stages E10.5, E.12.5, and E15.5. *CLSTN3* has not been associated with any disease phenotype so far.

##### *AKR1B10* (ALDO-KETO REDUCTASE FAMILY 1, MEMBER B10; OMIM *604707)

Individual 644_501 carried two compound heterozygous missense variants, c.121C>T (p.Arg41Trp) and c.124C>T (p.His42Tyr), in exon 2 of 10 (Ensembl GRChr37/hg19 transcript ENST00000359579.5) of *AKR1B10*. The protein consists of seven binding sites. Both variants are found just before the second nicotinamide adenine dinucleotide phosphate (NADP^+^) binding site (aa 44). Both variants were predicted to be deleterious by SIFT and probably damaging by PolyPhen-2, with CADD13_PHRED scores of 26.6 and 25.1. The aa are highly conserved down to Mm, Gg, Xt, and Dr (Arg and His). Our in-house murine transcriptome database shows expression during the embryonic stages E10.5, E.12.5, and E15.5, with an upregulation in E15.5 (log2FoldChange (15.5) of 1.44652099 and a *p*-value (15.5) of 1.3823 × 10^−22^, with a baseMean (15.5) of 775.040889. *AKR1B10* represents a human NADPH-dependent reductase belonging to the aldo-keto reductase (AKR) 1B subfamily. The enzyme is highly expressed in the epithelial cells of the stomach and intestine [33]. Interestingly, the disruption of NAD synthesis has previously been associated with congenital malformations in humans and mice [34]; however, *AKR1B10* has not been associated with any human disease phenotype so far.

##### *NDST4* (N-DEACETYLASE/N-SULFOTRANSFERASE 4; OMIM *615039)

Individual 828_501 carried two compound heterozygous missense variants, c.1600G>A (p.Val534Met) and c.778C>G (p.Leu260Val), in exons 7 and 2 of 14 (Ensembl GRChr37/hg19 transcript ENST00000264363.7) of *NDST4*. The variants are predicted to be deleterious and tolerated by SIFT and to be benign and possibly damaging by PolyPhen-2, with CADD13_PHRED scores of 23.4 and 22.7, respectively. The first aa is conserved down to Mm and the second variant is highly conserved down to Dr (Leu) for p.Leu260Val. Our in-house murine transcriptome database shows expression during the embryonic stages E10.5, E.12.5, and E15.5, with an upregulation in E15.5: log2FoldChange (15.5) of 3.99343455 and a *p*-value (15.5) of 1.3107 × 10^−12^, with a baseMean (15.5) of 30.375819.

Both variants reside within the first heparan sulfate N-deacetylase 4 region. So far, *NDST4* has not been associated with any human disease phenotype.

#### 3.1.4. Uniparental Disomy of *SVEP1*

##### *SVEP1* (SUSHI, VON WILLEBRAND FACTOR TYPE A, EGF, AND PENTRAXIN DOMAINS-CONTAINING 1; OMIM *611691)

Individual 181_501 carried a novel homozygous missense variant, c.5939C>T (p.Thr1980Ile), in exon 36 of 48 (Ensembl GRChr37/hg19 transcript ENST00000374469.6) of *SVEP1* (Table 4). The variant was predicted to be tolerated by SIFT and probably damaging by PolyPhen-2, with a CADD13_PHRED score of 23.8. The aa is evolutionarily conserved over Mm, Gg, Xt, and Dr (Thr). Our in-house murine transcriptome database shows the continuous expression of *Svep1* during the embryonic stages E10.5, E.12.5, and E15.5. *SVEP1* has not been associated with any human disease phenotype so far. A literature search revealed that the *Svep1* homozygous mutant embryos display multiple defects, such as edema, but also show abnormal development of the kidney and pelvis at E15.5 and E18.5 [35].

Overall, none of the variants found in our exome analysis can be classified as pathogenic, according to the ACMG classification criteria [36]. Hence, all variants that were prioritized and validated with Sanger sequencing should currently be interpreted as a variant of uncertain (or unknown) significance (VUS).

### 3.2. MIP Assay

To investigate the overall contribution of the above-mentioned candidate genes to BEEC, we re-sequenced those genes identified through exome analysis (*NR1H2*, *GKAP1*, *CLSTN3*, *AKR1B10*, *NDST4*, *PLEKHB1,* and *SVEP1*) and genes from the CBE phenocritical region, 22q11.2 (*LZTR1*, *SLC7A4*, *AIFM3*, *SNAP29*, *THAP7*, *P2RX6*, *CRKL)* in a cohort of 480 BEEC individuals. As outlined earlier, none of the individuals included in the MIP assay were included in our prior exome analysis.

Unfortunately, we did not identify any additional putative disease variants in the above-mentioned exome analysis-derived candidate genes. However, we identified two putative disease-causing variants in *LZTR1* and *SLC7A4* (Table 5). Sanger sequencing validated both variants. Due to a lack of paternal DNA and additional information such as family history, it remains unclear whether the variants did or did not occur de novo. Neither mother carried the respective variant. Both frameshift variants led to an early termination (Figure 1).

#### 3.2.1. *LZTR1* (LEUCINE ZIPPER-LIKE TRANSCRIPTIONAL REGULATOR 1; (OMIM *600574)

Individual 136_501 presented with CBE and carried a heterozygous frameshift variant, c.978_985del (p.Ser327ter6), in exon 9 (Ensembl GRChr37/hg19 transcript ENST00000215739.8) of *LZTR1*. The protein consists of six Kelch repeats and two BTB domains. The variant leads to a frameshift that affects the Kelch 5 (295aa-341aa) repeat and leads to an early termination. Our in-house murine transcriptome database shows continuous expression throughout the embryonic stages E10.5, E.12.5, and E15.5.

Dominant variants in *LZTR1* have been associated with Noonan syndrome 10 (OMIM #616564) and Noonan syndrome 2 (OMIM #605275) and susceptibility to Schwannomatosis (OMIM #615670). However, more interestingly, the deletion of *LZTR1* has been associated with the formation of congenital anomalies of the kidneys and urinary tract (CAKUT) [14,30,31,37,38]. *LZTR1* (c.978_985del, p.Ser327fster6) fulfills the ACMG criteria of pathogenicity (PVS1, PM2) [36]. However, this frameshift has previously been associated with Schwannomatosis and cardiovascular phenotypes (VCV001709123.4) and not with CBE, leaving some uncertainty about the involvement of this variant in CBE formation.

#### 3.2.2. *SLC7A4* (SOLUTE CARRIER 7 FAMILY 4; OMIM *603752)

Individual 458_501 presented with CBE and carried a novel heterozygous frameshift variant, c.1087delC (p.Arg363fster68), in exon 3 (Ensembl GRChr37/hg19 transcript ENST00000382932.2) of *SLC7A4*. The protein transcribes to 13 helical transmembrane domains. The variant is located between domains 7 (318-338aa) and 8 (365-385aa) and leads to an early termination. Our in-house murine transcriptome database shows the expression of *Slc7a4* throughout the time points E10.5, E.12.5, and E15.5, with an upregulation in E15.5: log2FoldChange (15.5) of 1685 and a *p*-value (15.5) of 7.4 × 10^−11^, with a baseMean (15.5) of 130.324411. *SLC7A4* has not been associated with any human disease phenotype so far. *SLC7A4* (c.1087delC, p.Arg363fster68) fulfills the ACMG criteria of a VUS in the current context (PM2) [36]. In particular, the lack of functional data and the lack of proof of a *de novo* occurrence of this variant suggests this variant to be VUS.

## 4. Discussion

Here, we used exome analysis with CE case-parent trios and large-scale re-sequencing for the identification of novel candidate genes and putative disease variants in the identified candidate genes. The exome survey in the CE case-parent trios identified two candidate genes harboring de novo variants (*NR1H2* and *GKAP1*), four candidate genes with autosomal-recessive biallelic variants (*AKR1B10*, *CLSTN3*, *NDST4*, and *PLEKHB1*), and one candidate gene with suggestive uniparental disomy (*SVEP1*). However, re-sequencing did not identify any additional variant carriers in these candidate genes.

Hitherto, 22q11.2 microduplication has been the only genetic risk factor that has been found to be significantly enriched among CBE individuals [1,14,30,31]. This finding prompted us to re-sequence *CRKL*, *LZTR1*, *THAP7*, *SLC7A4*, *AIFM3*, *SNAP29,* and *P2RX6*, which reside in the 22q11.2 phenocritical region. We thereby discovered two possible disease-causing frameshift variants that lead to early termination in *LZTR1* and *SLC7A4*.

In a male CBE individual, we identified a frameshift variant in *LZTR1* (c.978_985del, p.Ser327fster6). *LZTR1* is thought to be involved in a variety of inherited and acquired human disorders [39] and has been highlighted as a candidate gene for urogenital malformations [30,31,40,41]. *LZTR1* belongs to the BTB-Kelch superfamily, which play important roles during fundamental cellular processes, such as the regulation of gene expression, cell morphology, and migration [31], which are highly conserved during evolution [30]. Ubiquitous expression in mice (E9.5) has been shown in a previous study [30] and our in-house murine transcriptome database underlines these findings. This further reinforces *LZTR1* as a potential candidate gene for the BEEC phenotype.

Most recently, Lundin et al. found a novel variant (p.Ser698Phe) in *LZTR1* in one BEEC individual. Functional evaluation of the LZTR1 p.Ser698Phe variant in live NIH 3T3 cells showed that the concentration and cytoplasmic mobility differ between *Lztr1-wt* and *Lztr1-mut*, indicating the potential functional effect of *LZTR1-Mut* [31].

This information supports our finding and suggests *LZTR1* to be a strong candidate gene for CBE formation, warranting the functional characterization of all those *LZTR1* variants that have been described in association with CBE.

In a second male individual with CBE, we identified a frameshift variant in *SLC7A4* (c.1087delC, p.Arg363fster68). *SLC7A4* belongs to a family of cationic amino acid transporters. All exons of *SLC7A1*, *SLC7A2*, and *SLC7A4* are of similar or equal length; analysis of exon 3 of *SLC7A4* shows corresponding exons in *SLC7A1* and *SLC7A2*, which suggests that it may encode an important functional or regulatory domain [42]. Contrary to earlier reported findings of *Slc7a4* expression at E9.5 (the transcript was present but there was no expression) [30], the expression of *Slc7a4* in our in-house transcriptome database of uro-rectal tissue was continuous and upregulated during E15.5. While gains or losses in the chromosomal region 22q11.2, encompassing *SLC7A4* (OMIM *603752), have been associated with CAKUT or BEEC, no single base variant in *SLC7A4* has been associated with the human CAKUT or BEEC disease phenotypes so far.

## 5. Conclusions

An exome survey of case-parent trios with CE has identified novel candidate genes. These novel candidate genes require further investigation via larger re-sequencing analysis or functional in vitro or in vivo studies to support their involvement in the development of this disease before they can be annotated as BEEC-associated candidate genes. The re-sequencing of all genes residing in the CBE phenocritical region 22q11.2 has provided further support for *LZTR1* being implicated in CBE formation and suggests *SLC7A4* as a potential novel CBE candidate gene.

## Figures and Tables

**Figure 1 biomolecules-13-01117-f001:**
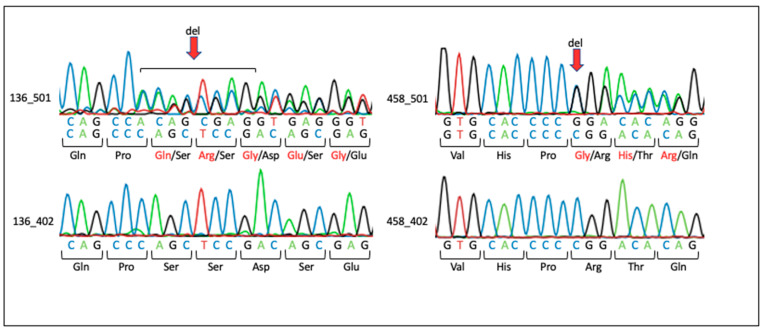
Sanger sequencing results for families with variants in *LZTR1* and *SLC7A4*. The red arrow indicates the position of base deletions. Amino acids in black show the wild type. Amino acids in red indicate changes due to deletion. Family 136: Individual 136_501 carries a heterozygous c.978_985del (p.Ser327fster6) deletion in *LZTR1*. The mother, individual 136_402, shows the wild-type sequence. Family 458: Individual 458_501 carries a heterozygous c.1087delC (p.Arg363fster68) deletion in *SLC7A4*. The mother, individual 458_402, shows the wild-type sequence.

**Table 1 biomolecules-13-01117-t001:** Molecular details and clinical features of individuals with autosomal-dominant monoallelic de novo variants.

Individual	420_501	
**Gene**	*GKAP1*	*NR1H2*
**GRCH37/hg19**	chr9:86383734	chr19:50882024
**Transcript**	NM_025211	NM_007121
**c.change**	c.737A>C	c.718C>T
**p.change**	p.Gln246Pro	p.Arg240Cys
**Variant consequence**	Missense	Missense
**Zygosity**	Heterozygous	Heterozygous
**Exon** ^1^	8/13	6/10
**MAF gnomAD v2.1** **(homozygotes)**	Not reported	Not reported
**Mode of Inheritance**	De novo	De novo
**SIFT (5.2.2)**	Deleterious (0.01)	Deleterious (0.01)
**PolyPhen-2 (2.2.2)**	Benign (0.011)	Probably damaging (0.931)
**CADD13_PHRED v1.6**	26.1	28.5
**Sex**	Female	
**Primary phenotype**	CE	

All information about the transcript refers to the canonical transcript. ^1^ Exon: the number before the slash indicates the affected exon. The number after the slash indicates the number of exons in this gene.

**Table 2 biomolecules-13-01117-t002:** Molecular details and clinical features of individual with autosomal-recessive biallelic homozygous variants.

Individual	420_501
**Gene**	*PLEKHB1*
**g.DNA position (GRCH37/hg19)**	chr11:73360114
**Transcript**	NM_021200
**c.change**	c.76G>A
**p.change**	p.Gly26Ser
**Variant consequence**	Missense
**Zygosity**	Homozygous
**Exon** ^1^	2/8
**MAF gnomAD v2.1**	0.0001189
**(homozygotes)**	(0)
**Mode of Inheritance**	Autosomal recessive
**SIFT (5.2.2)**	Deleterious (0)
**PolyPhen-2 (2.2.2)**	Probably damaging (1)
**CADD13_PHRED v1.6**	28.4
**Sex**	Female
**Primary phenotype**	CE

All information about the transcript refers to the canonical transcript. ^1^ Exon: the number before the slash indicates the affected exon. The number after the slash indicates the number of exons in this gene.

**Table 3 biomolecules-13-01117-t003:** Molecular details and clinical features of individuals with autosomal-recessive biallelic compound heterozygous variants.

Individual	390_501	644_501	828_501
**Gene**	*CLSTN3*	*AKR1B10*	*NDST4*
**g.DNA position (GRCH37/hg19)**	chr12:7303179 and chr12:7310183	chr7:134215449 and chr7:134215452	chr4:115792043 and chr4:115997415
**Transcript**	NM_014718	NM_020299	NM_022569
**c.change**	c.2285A>T and c.2626C>T	c.121C>T and c.124C>T	c.1600G>A and c.778C>G
**p.change**	p.Gln762Leu and p.Arg876Cys	p.Arg41Trp and p.His42Tyr	p.Val534Met and p.Leu260Val
**Variant consequence**	Missense and Missense	Missense and Missense	Missense and Missense
**Zygosity**	Heterozygous	Heterozygous	Heterozygous
**Exon** ^1^	15/18 and 17/18	2/10 and 2/10	7/14 and 2/14
**MAF gnomAD v2.1 ** **(homozygotes)**	Not reported and 0.00004277 (0)	0.0001026 (0) and Not reported	0.0001562 (0) and 0.00005672 (0)
**Mode of Inheritance**	Compound Heterozygous	Compound Heterozygous	Compound Heterozygous
**SIFT (5.2.2) ** **PolyPhen-2 (2.2.2) ** **CADD13_PHRED v1.6**	Tolerated (0.54) and deleterious (0) Benign (0.015) and possibly damaging (0.841) 20.2 and 25.3	Deleterious (0) and deleterious (0) Probably damaging (0.987) and probably damaging (0.978) 26.6 and 25.1	Deleterious (0.02) and tolerated (0.2) Benign (0.083) and possibly damaging (0.506) 23.4 and 22.7
**Sex**	Male	Female	Male
**Primary phenotype**	CE	CE	CE

All information about the transcript refers to the canonical transcript. ^1^ Exon: the number before the slash indicates the affected exon. The number after the slash indicates the number of exons in this gene.

**Table 4 biomolecules-13-01117-t004:** Molecular details and clinical features of an individual with a uniparental isodisomy variant.

Individual	181_501
**Gene**	*SVEP1*
**g.DNA position (GRCH37/hg19)**	chr9:113189907
**Transcript**	NM_153366
**c.change**	c.5939C>T
**p.change**	p.Thr1980Ile
**Variant consequence**	Missense
**Zygosity**	Homozygous
**Exon** ^1^	36/48
**MAF gnomAD v2.1**	Not reported
**Mode of Inheritance**	Uniparental disomy
**SIFT (5.2.2)**	Tolerated (0.1)
**PolyPhen-2 (2.2.2)**	Probably damaging (0.985)
**CADD13_PHRED v1.6**	23.8
**Sex**	Male
**Age of onset**	Congenital
**Primary phenotype**	CE

All information about the transcript refers to the canonical transcript. ^1^ Exon: the number before the slash indicates the affected exon. The number after the slash indicates the number of exons in this gene.

**Table 5 biomolecules-13-01117-t005:** Molecular details and clinical features of individuals with variants in *LZTR1* and *SLC7A4*.

Individual	136_501	458_501
**Gene**	*LZTR1*	*SLC7A4*
**g.DNA position (GRCH37/hg19)**	chr22:21346103-21346110	chr22:21384536
**Transcript**	NM_006767	NM_004173
**c.change**	c.978_985del	c.1087delC
**p.change**	p.Ser327ter6	p.Arg363ter68
**Variant consequence**	Frameshift	Frameshift
**Zygosity**	Heterozygous	Heterozygous
**Exon** ^1^	9/21	3/5
**gnomAD MAF**	Not reported	Not reported
**Mode of Inheritance**	N/A	N/A
**Sex**	Male	Male
**Primary phenotype**	CBE	CBE

All information about the transcript refers to the canonical transcript. ^1^ Exon: the number before the slash indicates the affected exon. The number after the slash indicates the number of exons in this gene.

## Data Availability

The data that support the findings of this study are available on request from the corresponding author. The data are not publicly available due to privacy or ethical restrictions.

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
