# Peer review of "Exome Survey and Candidate Gene Re-Sequencing Identifies Novel Exstrophy Candidate Genes and Implicates LZTR1 in Disease Formation"

_biomolecules, 2023, doi:10.3390/biom13071117_

Round 1

Reviewer 1 Report

In this paper by Köllges et al., the authors utilized an exome survey for the bladder exstrophy-epispadias complex (BEEC) cohort and identified candidate genes and variants that might be disease-related. They further implicated LZTR1 in the disease formation of classic bladder exstrophy (CBE). Overall, it is an interesting topic for research on BEEC, especially for the molecular diagnoses of patients with BEEC. However, there are a few issues that need to be addressed.

Major comments:

·      Although the authors listed the best candidates and related information to the variants that the authors could find, e.g. MAF gnomAD, conservation of amino acids and bases, in silico prediction tools, et al., there’s no standard interpretation for any of the variants discovered in the study, nor any functional assays to show pathogenicity of variants at the levels of protein or RNA. It is strongly suggested that the authors use ACMG/AMP Codes for the interpretations of novel variants (Reference PMID: 25741868).

·      If possible, please add segregation data for the variants of LZTR1 (c.978_985del, p.Ser327fster6) and SLC7A4 (c.1087delC, p.Arg363fster68), which is a very essential part to molecularly diagnose a patient with inherited diseases.

·      Functional assays for any of the variants discovered in this study are encouraged.

·      Please submit the novel variants and classification to public variant databases like ClinVar.

Other comments:

·      Please add the symbols preceding an OMIM number, for example, an asterisk (*) before a gene, a number symbol (#) before a phenotype whose molecular basis is known, or a percent sign (%) before a confirmed mendelian phenotype or phenotypic locus for which the underlying molecular basis is not known, BEEC belongs to this category.

·      Line 105-118, please add references for the tools.  

·      How was the quality control for the Exome Sequencing? How deep was the coverage?

·      Line 152, "Mm." should be "Mm".

·      In tables, for “Exon”, please indicate what the numbers before and after “/” mean.

·      Please give reference scores when using the tools SIFT (5.2.2), PolyPhen-2 (2.2.2), and CADD13_PHRED v1.6.

·      In tables, for “Inheritance”, was this to describe the variant or the disease? Please make it clear and unify the format. Currently, zygosity and inheritance are mixed up somewhere.

·      Variants of neither LZTR1 nor SLC7A4 were correctly named, they are both frameshift variants instead of nonsense variants. A nonsense variant, is a single DNA nucleotide that has been altered such that an amino acid codon becomes a stop codon, leading to truncation of the encoded protein. Please see the definition referring to this page <https://illustrated-glossary.nejm.org/term/nonsense_variant>

·      In Figure 1, what are the top and bottom sequences? What do the red and black amino acids mean? It needs to be labeled in the figure or noted in the figure legend. Please avoid the wavy red underlines generated by automatic spelling checking.

·      Are the fathers in family A and B healthy? Do they have family histories?

·      Why during the first round of exome sequencing, variants in LZTR1 and SLC7A4 were not identified?

·      OMIM number should be put right after the gene or the phenotype, line 155 and line 332 are two examples.

·      Line 165, the OMIM number for gene GKAP1 should be *611356, instead of 6111356.

·      3.2. MIP” is suggested to be changed into “3.2. MIP assay” to avoid looking like a gene symbol.

·      Line 330, “gain or losses”, change it to “gain or loss” or “gains or losses”.

The overall English writing is good, and the flow is easy to follow. Please check through the whole manuscript to see if they're more typos or mistakes.

Reviewer 2 Report

The Authors present a report on exome analysis of case-parent trios with cloacal exstrophy (CE), which is the most severe form of the BEEC (Bladder Exstrophy-Epispadias Complex). Furthermore, they conducted a survey of the exome of a sib-pair with classic bladder exstrophy (CBE) and epispadias only. Finally, the authors performed large-scale re-sequencing of individuals with CBE to identify novel candidate genes identified through the present exome analysis and previously reported candidate genes within the CBE phenocritical region 22q11.2.

Through the exome survey in CE case-parent trios, three candidate genes harboring de novo variants (FRS3, NR1H2, GKAP1), four candidate genes with autosomal-recessive biallelic variants (AKR1B10, CLSTN3, NDST4, PLEKHB1) and one candidate gene with suggestive uniparental disomy (SVEP1) have been identified. However, the analysis of the affected sib-pair reveled no candidate gene. Re-sequencing of genes within the 22q11.2 CBE phenocritical region identified two highly conserved novel nonsense variants in two independent CBE males in LZTR1 (c.978_985del, p.Ser327fster6) and in SLC7A4 (c.1087delC, p.Arg363fster68).

Based on the results, the present study further implicates LZTR1 in CBE formation. Furthermore, the candidate genes derived from exome analysis of CE individuals may not represent a frequent cause for other BEEC phenotypes but justify molecular analysis.

In my opinion, this manuscript is of high quality and is highly interesting.  Thus, I recommend its acceptance with minor corrections. I have some suggestions to improve it.

Minor correction

Conclusion (Abstract): According to previous studies, our study further implicates LZTR1 in CBE formation.

Exome analysis derived candidate genes from CE individuals may not represent a frequent cause

for other BEEC phenotypes and warrant molecular analysis before implication in the disease for

mation can be assumed

·                     Conclusion sentence should be rewritten in more clear and direct way. 

Suggestions

          Pls. for Interpretation and Classifying of Sequence Variants use: Richards S, Aziz N, Bale S, Bick D, Das S, Gastier-Foster J, Grody WW, Hegde M, Lyon E, Spector E, Voelkerding K, Rehm HL; ACMG Laboratory Quality Assurance Committee. Standards and guidelines for the interpretation of sequence variants: a joint consensus recommendation of the American College of Medical Genetics and Genomics and the Association for Molecular Pathology. Genet Med. 2015 May;17(5):405-24. doi: 10.1038/gim.2015.30. Epub 2015 Mar 5. PMID: 25741868; PMCID: PMC4544753.

          Pls. for all genes (i.e FRS3) report extension name of gene and OMIM number (i.e.  FIBROBLAST GROWTH FACTOR RECEPTOR SUBSTRATE 3; MIM 607744)………….

Reviewer 3 Report

The bladder exstrophy-epispadias complex (BEEC) is a rare congenital birth defect without clear knowing genetic etiology. The authors of this article tried to identify potential candidate genes for this congenital anomaly. They initially sequenced 14 trios and one sib-pair by whole exome sequencing. Eight candidate genes were identified. they further re-sequenced those genes and other 7 genes from the 22q11.2 region in a cohort of 480 BEEC patients. Unfortunately, none of eight genes were re-identified from this cohort. However, they found a nonsense LZTR1 variant from the cohort. Another two LZTR1 variants have been previously identified from BEEC patients. The authors concluded that LZTR1 might be implicated in CEB formation. 

Major Concerns : Although a few papers have reported that 22q11.2 microduplication is associated with CBE, claiming a gene with pathogenic variant in a microduplication region as a candidate gene does not make much sense for me. I would more believe it if it were a microdeletion.  secondly, LZTR1 has been associated with AD-Noonan syndrome, AR-Noonan syndrome, and Schwannomatosis. All with incomplete penetrance. Some of pathogenic LZTR1 variants present in gnomAD at higher than rare level. For example, the pathogenic nonsense variant LZTR1:c.628C>T, which was identified in one CEB patient, has an allele frequency of 0.010878% in Non-finish Europeans. It is hard to believe only one variant was identified in a 480 BEEC cohort, if it were a candidate gene for BEEC. Lastly, for those congenital anomalies involving muti-organ system, structure variant that involves many genes has higher chance to be the culprit than a variant in a single gene. Whole exome sequencing is not perfect for structure variant calling, but I would recommend to  check if there are any structure variants could be identified in this cohort.

Others:

1, The variant FRS3:c.874G>A has an allele frequency of 0.000076657in Non-Finnish Europeans and 0.000050661 in all populations. This is higher than the cutoff described in the methods, which is 0.0001.

2, At Page 4 line 152, it says the amino acid in Dr is Arg, however when I check it, it is a Gly. I do not see any species have an Arg at this position.

3, The variant GKAP1:c.737A>C is located at the penultimate position of exon 8, which is very close to the canonical splicing donor site. SpliceAI predicts a high chance of donor loss for this variant. So it might not be a missense variant but a splicing variant. This should be discussed in the manuscript.

4, the LZTR1”c.978_985del has been reported 3 times in ClinVar as pathogenic should be mentioned in the manuscript.

Round 2

Reviewer 1 Report

Thanks to the authors for making changes accordingly! Some of my suggestions have been adequately addressed, although some other points haven’t been revised appropriately.

1. Please submit a version of the revised paper with ALL edits highlighted, including changes and deletions. What happened to FRS3? Why it disappeared? The authors made some changes but did not mark them nor elaborate the reason for the changes. It’s time-consuming and not appropriate for the reviewers to go back and forth to the first version and try to find what was modified.

2. Please add details of the ACMG/AMP codes for variant interpretation to the manuscript, not just the codes. The authors made mistakes when using ACMG/AMP codes for the LZTR1 (c.978_985del, p.Ser327fster6) and SLC7A4 (c.1087delC, p.Arg363fster68), for example, PM4 is used for in-frame variants. Please refer to PMID: 25741868 and make corrections. Regarding other variants reported in this paper, most are VUS, but it’s still better to interpret them than just list them.

3. The cohort not only lacks paternal DNA but also lacks ANY paternal information, including the family history part, please add the clarification into the manuscript, so to reduce readers’ confusion.

4. “Number behind the slash indicates number of exons in this gene (canonical transcript).“ Are the transcripts you put in the chart all canonical transcripts? Please make it clear in the manuscript.

5. “As requested by the reviewer, we will submit the variants to ClinVar.” Please put the accession ID in the manuscript. Could the authors add all the variants reported in this paper to ClinVar, even if they are VUS?

6. Please add into the manuscript: the average exome read coverage, the percent of targeted bases over 20× coverage (exome sequencing generally requires a minimum coverage of 20x to achieve a 95% on-target single-nucleotide polymorphism detection sensitivity, PMID: 25973577, 29789557, 25038816), as well as the minimal coverage for variants to be eligible for further prioritization as the authors addressed in the letter.

7. For my question “why during the first round of exome sequencing, variants in LZTR1 and SLC7A4 were not identified?” It was for the authors to put into discussion. The goal of publications is for the readers to learn something new, a new variant can be it, lessons the authors learned from mistakes or negligence are also treasurable. That’s also the significance of doing re-sequencing in this project.

8. “OMIM number should be put right after the gene or the phenotype, line 155 and line 332

are two examples.” This hasn’t been corrected. Line 332, OMIM *603752 is for SLC7A4, not for BEEC disease phenotype, so it needs to be right after SLC7A4, instead of after BEEC disease phenotype.

Reviewer 3 Report

NO

Author Response

There were no further comments. Thank you.